

# WHONDRS-GUI: a web application for global survey of surface water metabolites

Xinming Lin,  Huiying Ren,  Amy E. Goldman,  James C. Stegen and Timothy D. Scheibe

Pacific Northwest National Laboratory, Richland, WA, United States of America

## ABSTRACT

**Background**. The Worldwide Hydrobiogeochemistry Observation Network for Dynamic River Systems (WHONDRS) is a consortium that aims to understand complex hydrologic, biogeochemical, and microbial connections within river corridors experiencing perturbations such as dam operations, floods, and droughts. For one ongoing WHONDRS sampling campaign, surface water metabolite and microbiome samples are collected through a global survey to generate knowledge across diverse river corridors. Metabolomics analysis and a suite of geochemical analyses have been performed for collected samples through the Environmental Molecular Sciences Laboratory (EMSL). The obtained knowledge and data package inform mechanistic and data-driven models to enhance predictions of outcomes of hydrologic perturbations and watershed function, one of the most critical components in model-data integration. To support efforts of the multi-domain integration and make the ever-growing data package more accessible for researchers across the world, a Shiny/R Graphical User Interface (GUI) called WHONDRS-GUI was created.

**Results**. The web application can be run on any modern web browser without any programming or operational system requirements, thus providing an open, well-structured, discoverable dataset for WHONDRS. Together with a context-aware dynamic user interface, the WHONDRS-GUI has functionality for searching, compiling, integrating, visualizing and exporting different data types that can easily be used by the community. The web application and data package are available at https://data.ess-dive.lbl.gov/view/doi:10.15485/1484811, which enables users to simultaneously obtain access to the data and code and to subsequently run the web app locally. The WHONDRS-GUI is also available for online use at Shiny Server (https://xmlin.shinyapps.io/whondrs/).

## INTRODUCTION

Watershed systems play a critical role in human society and environmental health by providing ecosystem services such as drinking water for humans, water for irrigation and industrial activities, and habitats for plants and animals (*Postel & Thompson Jr, 2005*). The

Corresponding author
James C. Stegen,
James.Stegen@pnnl.gov

associated river corridors, including the river, adjacent land, and subsurface environments, are key components of watershed systems (*Harvey & Gooseff, 2015*). Recurring, episodic, or chronic perturbations to river corridors, such as those caused by dam operations, wastewater effluent, floods, storm pulses, and droughts, are common but difficult to study due to their transient nature. Understanding the connections among hydrologic, biogeochemical, and microbial function within river corridors is not only important to help predict the outcomes of these perturbations in terms of watershed function, but also to maintain a healthy environment (*Graham et al., 2019*). The Worldwide Hydrobiogeochemistry Observation Network for Dynamic River Systems (WHONDRS) (*Stegen & Goldman, 2018*) is a global consortium of researchers and other interested parties dedicated to studying these connections.

WHONDRS generates knowledge across river corridors from local to global scales. For one ongoing study, surface water metabolite and microbiome samples are collected by local collaborators around the world through a global survey. This study aims to provide broad understanding of the functioning of dynamic river corridors. Metabolomics analysis via Fourier transform ion cyclotron resonance mass spectrometry (FTICR-MS) (*Marshall, Hendrickson & Jackson, 1998*) and a supporting suite of geochemical analyses have been conducted through the Environmental Molecular Sciences Laboratory (EMSL) for the field-collected samples. The data obtained are intended to inform mechanistic and data-driven models to enhance predictions of the hydro-biogeochemical function of river corridors and watersheds, thus serving as a critical component of the model-data integration process (*Scheibe et al., 2018*). However, as the number of collected samples increases, the resulting WHONDRS data package will reach a size and complexity that makes it challenging for researchers to manage and search for the part of the data package in which they are most interested.

To support the efforts of multi-domain integration of physical, biological, and chemical processes within river corridors, a shiny-based web application called WHONDRS-GUI was created. This web app is intended to make the collected, ever-growing WHONDRS data package more accessible for researchers to use in performing new analyses or updating prior analyses, in keeping with the FAIR principles for scientific data management (*Wilkinson, 2016*).

The WHONDRS-GUI is a user-friendly graphical user interface (GUI) developed using R (*R Development Core Team, 2018*), and Shiny (*Winston Chang et al., 2018*). Like any other Shiny app, the WHONDRS-GUI can run on any modern web browser without any programming or operational system requirements. It provides open, well-structured, discoverable data, visualization of specific data types, as well as a global map of all collected sampling sites. Together with a context-aware dynamic user interface, the WHONDRS-GUI includes functionality for searching, compiling, integrating, and exporting different data types that can easily be used by the scientific community. The data types used to build WHONDRS-GUI include metadata of collected surface water samples, FTICR data and geochemistry data from analyses performed in EMSL, surface water hydrographs collected from existing instrumentation, and standardized photos of each field system (*Stegen et al., 2018*). By engaging with the global scientific community, the WHONDRS-GUI promotes

sharing and extracting knowledge from the WHONDRS data package (*Stegen et al., 2018*), and enables researchers to better address science questions relevant to their individual sites as well as perform integrative cross-site analyses.

## MATERIALS & METHODS

### Data sources and description

The data package used in this study is from an on-going global survey of surface water metabolites that started in 2018. Surface water metabolites and microbiome samples from streams and river corridors around the world were collected by local collaborators and sent to EMSL for analysis. Metabolomics analysis via FTICR-MS and a suite of geochemical analyses were conducted on all samples at EMSL. The application is designed to use five different data types from the data package:

1. Metadata of collected surface water samples. These are recorded by local collaborators who collected the surface water samples. The metadata include names of samplers, sample ID, sampling time, location, water temperature, distance from gauging station, and river names of the water samples.
2. FTICR-MS data. These are results of metabolomic analysis for collected surface water samples.
3. Basic geochemistry data. These are results of geochemistry analyses conducted for surface water samples, including ion and non-purgeable organic carbon concentrations.
4. Surface water hydrographs. These are open-access water-resources data of gauging stations downloaded from United States Geological Survey (USGS) website based on sampling locations of collected surface water samples.
5. Standardized photos taken from each field system. These are photos of field systems taken by collaborators.

Following data cleaning, manipulation and adjustment of raw data (performed using the 'dplyr' R package (*Wickham et al., 2015*)), all available data types, except photos, were saved in comma-separated values (CSV) file format. The processed data types were then used as inputs for building the WHONDRS-GUI. The WHONDRS-GUI provides the option of downloading all or part of the data package for users who are interested in performing their own analysis.

### Graphical user interface

The WHONDRS-GUI interface was created in R console using Shiny and its supplementary packages. It is an interactive web application that can be run on a local machine using R or R studio. The web application is also deployed online by hosting to Shiny Server. This allows for fast access to and easy sharing of the web application through any modern web browser.

The WHONDRS-GUI interface is organized into dashboard pages using the 'shinydashboard' package (*Chang & Borges Ribeiro, 2017*). It utilizes a tab-based dashboard format to make accessing each data type fast and intuitive. Data types, such as 'Metadata', 'FTICR', 'Geochemistry', 'Hydrograph' and 'Photo', are displayed with their names listed on the left-hand sidebar of the interface. The output tables, maps, and plots are displayed

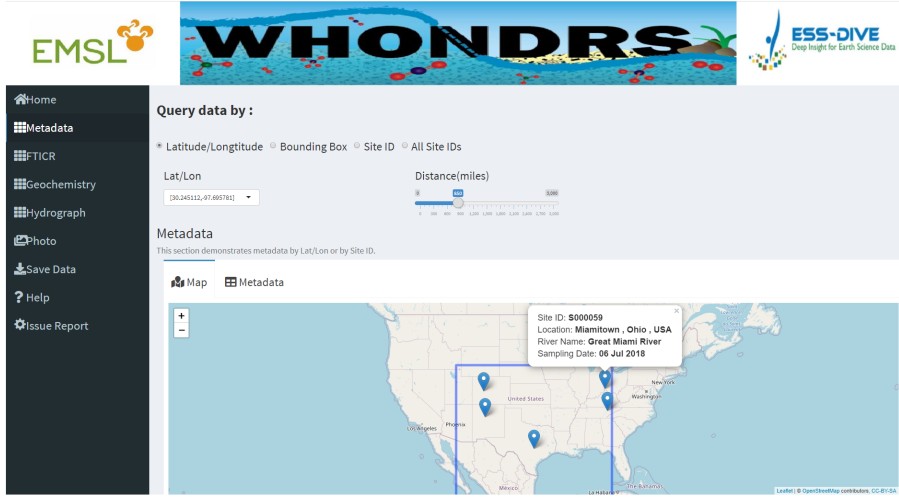

**Figure 1 Metadata dashboard page.** The 'Metadata' dashboard page shows a global map of collected sampling sites. User input widgets on the top of the page, including 'Latitude/Longtitude', 'Bounding Box', 'Site ID', or 'All Stie IDs', allow users to search for a subset of data based on different options. Clicking a marker on the map will return the metadata of the selected site.

to the right of the sidebar, as the main body of the dashboard page. Users can explore each data type by clicking tab names on the left. This will result in the corresponding output page appearing on the right.

The first tab on the left-hand sidebar of the interface, and the landing dashboard page when the web application is first loaded, is the 'Home' tab. Clicking this tab will lead users to the homepage of the web application. This page gives a short description of the WHONDRS-GUI. Clicking the 'GET STARTED' action button below the description paragraph will lead users to the 'Help' dashboard page, where users can find more information about the web application and how to navigate it locally.

Following the 'Home' dashboard page are dashboards pages for the five available data types, including 'Metadata', 'FTICR', 'Geochemistry', 'Hydrograph' and 'Photo'. User input widgets on the top of these dashboard pages, allow users to search for a subset of data based on different options. As shown in Fig. 1, these options include selecting latitude and longitude of a specific site, selecting site IDs, or dragging a bounding box on the map. Each option requires different inputs as searching criteria. For example, there are two inputs for the latitude/longitude option. One is the latitude and longitude values of a site, which can be chosen directly from the 'Lat/Lon' input widget. The other input is the distance value from a selected site, which can be obtained by dragging the slider of the 'Distance(miles)' widget.

Dashboard pages for all data types except 'Photo' display a well-structured data table for selected sites. For the 'Metadata' dashboard page, it also displays a global map of collected sampling sites. Clicking a marker on the map will return the metadata of the selected site, which includes site ID, location, river name, and sampling date, as shown in Fig. 1. Apart from displaying data tables for a selected site, the 'Hydrograph' dashboard page displays

a time-series plot of discharge and/or gage height for the selected site. This allows users to visually assess the quality of the timeseries data and see how gage height and discharge change over time for the site that they are interested in. Dragging the slider of the 'Time range (days)' widget allow users to change the time period of discharge and gage height.

In the 'Photo' dashboard page, four standardized photos of a selected field system are exhibited. This allows users to know the physical environment where the surface water samples were collected. The previously selected subset of data can be exported to a separate folder in the default directory by clicking the 'Save Data' button in the selected dashboard. Users can then use the exported subset of data locally for their research needs.

### Availability

The WHONDRS-GUI is available through Shiny Server (https://xmlin.shinyapps.io/whondrs/) for online use, which allows users to get access to the web application through any modern web browser without installing R or R studio. The website is free and open to all users without login requirement. The WHONDRS data package as well as the R code are hosted on the ESS-DIVE data archive (https://data.ess-dive.lbl.gov/view/doi:10.15485/1484811). This enables users to simultaneously obtain access to the data and code and to subsequently launch the web app locally from any R environment (e.g., R and RStudio) without any programming or operational system requirements. Documentation for how to run the WHONDRS-GUI locally can be found in the downloaded package.

## RESULTS AND DISCUSSION

The WHONDRS-GUI, a web application written in R programming language, has been successfully implemented to assist in searching, compiling, visualizing, integrating and exporting different data types from the WHONDRS data package. It makes the ever-growing WHONDRS data package more accessible for researchers to use in performing new analyses or updating prior analyses. As an interactive web application, the WHONDRS-GUI is simple and intuitive to use without the need to download and install any specialized software.

The WHONDRS-GUI web application provides interactive data tables, global maps, timeseries plots and standardized river photos of water samples collected across the world. Five data types, which include 'Metadata', 'FTICR', 'Geochemistry', 'Hydrograph' and 'Photo', are being used to demonstrate the collected WHONDRS data. Data processing, subsetting and manipulating are done automatically based on user inputs, such as latitude/longitude, site ID and distance.

Useful and detailed metadata of the collected samples are available on the 'Metadata' dashboard page. These metadata include site ID, sampling time, location, and river names of the water samples etc. A global map interactively displaying physical locations of selected samples based on user inputs is also available on the same dashboard page (Fig. 1). The global map will automatically zoom in or zoom out according to the number of selected site IDs.

Data tables for all data types except 'Photo' are generated for selected sample sites on corresponding dashboard pages. Figure 2 shows a preview of the geochemistry data

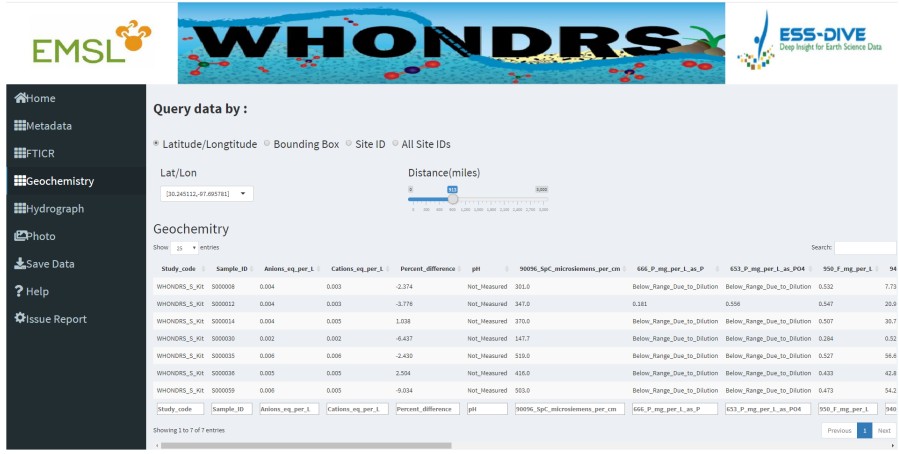

**Figure 2** **Geochemistry dashboard page.** A preview of the geochemistry data table for seven selected sites.

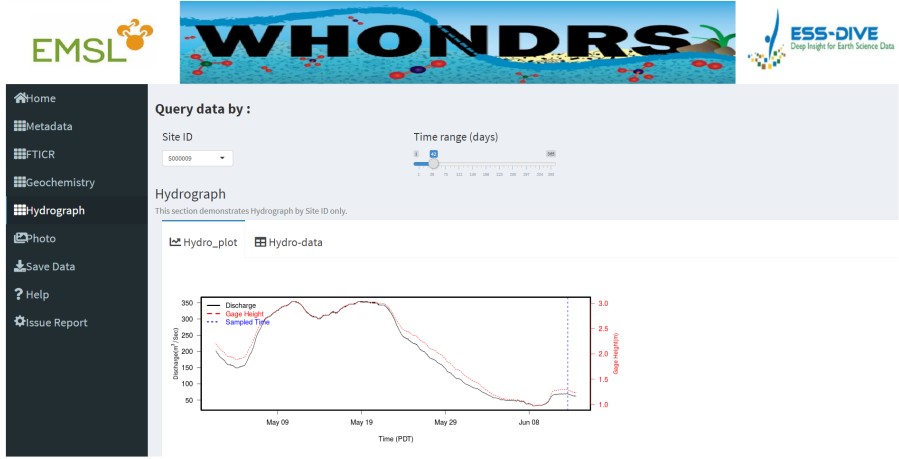

**Figure 3** **Hydrograph dashboard page.** This dashboard page displays an interactive time-series plot of discharge and/or gage height for a selected site. The plot shown in this fugure is a time-series plot of discharge and gage height for site 'S000009'.

table for seven selected sites in the 'Geochemistry' dashboard page. For the analysis of hydrograph data, depending on the chosen sample ID, a plot of discharge and/or gage height is generated showing the changing pattern of discharge and/or gage height over time, as shown in Fig. 3. The time-series plot is a variable width plot that defaults to show a time range of 30 days start from the earliest day of the available data. The time range can be adjusted between 1 day to 365 days using the slider of the 'Time range (days)' widget (Fig. 3).

Apart from the functionality of the WHONDRS-GUI demonstrated above, it also provides the option for users who are interested in performing further analyses of the data to download the whole data package or part of it, and to save to local drive, by clicking
the 'Save Data' button on the dashboard page. If a user identifies a bug or problem while running the WHONDRS-GUI, it can be reported under the "Issue Report" section. The bug report form is hosted on Google Drive and records each bug report and transmits it to the WHONDRS team automatically.

The results described above show that the WHONDRS-GUI provides simple and convenient tools for researchers with limited programming experience to access and manipulate the WHONDRS data, and to interactively visualize some of the data types. The web application also offers a solution to integrate large amount of information contained in WHONDRS data with an intuitive and user-friendly interface, which further facilitates the use and exploration of WHONDRS data by the wider scientific community. The current version of the web application is focused on the ability to quickly demonstrate available data types, with limited ability to perform data analysis. Also, the format of input and output data used by the web application is limited to CSV file. To improve the flexibility of the WHONDRS-GUI and its ability to conduct more advanced data analysis, we will continue to develop this web application and implement these user-friendly changes in future versions.

## CONCLUSIONS

WHONDRS data are being collected to understand multi-domain connections within river corridors and help predict watershed function. In this work, we developed a web application, WHONDRS-GUI, to make WHONDRS data findable, accessible, interoperable, and reusable (*Wilkinson et al., 2016*) and to encourage users to perform their own analyses and extract new information from the data. The WHONDRS-GUI provides a convenient and user-friendly interface for different WHONDRS data types. It allows users to dynamically interact with the data package. Researchers can utilize the WHONDRS-GUI to find specific sample data.

The WHONDRS-GUI is an important element of understanding connections among hydrologic, biogeochemical, and microbial function within river corridors. The GUI alone will not provide this understanding, however, and must be coupled with additional analyses and tools as well as additional data types that are not currently in the datasets. For example, detailed molecular properties of dissolved organic matter (DOM) are provided by the FTICR data, which can be paired with data on environmental characteristics (e.g., vegetation type, land use, ecosystem productivity, etc.) that are publicly available. Because all WHONDRS data are georeferenced, these additional environmental data can be extracted from other sources and paired with WHONDRS data to evaluate aspects of the environment that explain variation in DOM chemistry at the global scale. Outcomes of such analyses would provide novel hypotheses to be evaluated through additional sampling, experimentation, and/or modeling.

The data can also be more directly analyzed to understand spatial and temporal variation in biogeochemically-relevant properties, such as the thermodynamic favorability of DOM for microbial oxidation. For example, (*Danczak et al., 2020a*; *Danczak et al., 2020b*) recently discovered the principle of 'thermodynamic redundancy,' whereby compositionally

distinct metabolomes exhibit consistent levels of thermodynamic favorability. This new principle was revealed, in part, by linking FTICR data to a new conceptual/theoretical framework derived from meta-community ecology (*Danczak et al., 2020a*; *Danczak et al., 2020b*). This framework is referred to as 'meta-metabolome ecology' and provides a new window into river corridor function by integrating ecological principles with ultrahigh resolution molecular properties of DOM chemistry. The data necessary to use tools/concepts from meta-metabolome ecology can be accessed via the WHONDRS-GUI.

To provide additional opportunities to generate new knowledge of integrated river corridor function, WHONDRS is collaborating with the Joint Genome Institute (JGI) to generate metagenomic and metatranscriptomic data of microbial communities. These data can be used with the FTICR and geochemical data (from the WHONDRS-GUI) to inform biogeochemical reaction network models that can be further integrated with reactive transport codes that simulate reactions and the movement of water and other materials (e.g., nutrients) through river corridors. The develop of such an integrated modeling framework is nascent, and some of the critical theory and modeling tools have recently been developed (*Song et al., 2020*). There are significant efforts underway to implement this integrated framework via publicly available tools in the US Department of Energy Systems Biology Knowledgebase (KBase; https://kbase.us/). Once fully implemented, researchers will be able to discover and access WHONDRS data using the GUI and other tools, and then integrate/study these data using tools in KBase that generate and link metabolic, biogeochemical reaction, and reactive transport models. This offers myriad opportunities to generate new knowledge, such as how hydrologic perturbation impacts biogeochemical function via changes to microbial metabolism. Researchers could use numerical *in silico* experimentation across places and times sampled by WHONDRS to study such impacts and connections. The WHONDRS-GUI is a key piece of the cyberinfrastructure needed to enable such studies and the broader vision of understanding connections among hydrologic, biogeochemical, and microbial function within river corridors.

Both the data package and the R code are hosted on the ESS-DIVE data archive (https://data.ess-dive.lbl.gov/view/doi:10.15485/1484811), which enables users to simultaneously obtain access to the data and code and to subsequently run the web app locally. The WHONDRS-GUI is also available for online use at Shiny Server (https://xmlin.shinyapps.io/whondrs/).

## ACKNOWLEDGEMENTS

We would like to thank all collaborators for collecting the surface water samples and the EMSL teams for conducting a series of metabolomics analysis and geochemical analyses.

### Funding

This research is supported by the U.S. Department of Energy (DOE), Office of Biological and Environmental Research (BER), as part of the Subsurface Biogeochemical Research

(SBR) Scientific Focus Area (SFA) at the Pacific Northwest National Laboratory (PNNL). PNNL is operated by Battelle for DOE under Contract DE-AC06-76RLO1830. The funders had no role in study design, data collection and analysis, decision to publish, or preparation of the manuscript.

## Grant Disclosures

The following grant information was disclosed by the authors:
U.S. Department of Energy (DOE).
Office of Biological and Environmental Research (BER).
Scientific Focus Area (SFA) at the Pacific Northwest National Laboratory (PNNL): DE-AC06-76RLO1830.

## Competing Interests

Timothy D. Scheibe is an Academic Editor for PeerJ.

## Author Contributions

- Xinming Lin conceived and designed the experiments, performed the experiments, analyzed the data, prepared figures and/or tables, authored or reviewed drafts of the paper, and approved the final draft.
- Huiying Ren and Amy E. Goldman conceived and designed the experiments, performed the experiments, analyzed the data, authored or reviewed drafts of the paper, and approved the final draft.
- James C. Stegen and Timothy D. Scheibe conceived and designed the experiments, performed the experiments, authored or reviewed drafts of the paper, and approved the final draft.

## Data Availability

The codes and data package are available at:

Stegen JC; Goldman AE; Blackburn S E ; Chu RK ; Danczak RE; Garayburu-Caruso VA; Graham EB; Grieshauber C; Lin X; Morad JW; Ren H; Renteria L; Resch CT; Tfaily M; Tolic N ; Toyoda JG ; Wells JR ; Znotinas KR (2018): WHONDRS Surface Water Sampling for Metabolite Biogeography. Worldwide Hydrobiogeochemistry Observation Network for Dynamic River Systems (WHONDRS). DOI: https://data.ess-dive.lbl.gov/view/doi:10.15485/1484811.

The web application is available for online use at Shiny Server https://xmlin.shinyapps.io/whondrs/).

## Supplemental Information

Supplemental information for this article can be found online at http://dx.doi.org/10.7717/peerj.9277#supplemental-information.

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
