# Peer review of "WHONDRS-GUI: a web application for global survey of surface water metabolites"

_PeerJ, doi:10.7717/peerj.9277_

## Round 0.1 · original submission · Minor Revisions

Please consider the suggestions made by both the reviewers and resubmit a revised version. Please make sure that you correctly identify the changes made, if any. If you disagree with any of the reviewers' comments please give sufficient explanations as well.

Reviewer 1 ·

Basic reporting

The figures appear to be screenshots of various pages of the WHONDRS-GUI from within RStudio. They contain ill-defined edges consistent with screenshots. Two of the figures still show the breadcrumb bar, which is not appropriate for a publication-quality figure. Lastly, the screenshots are not of sufficient resolution for publication. Thus, I recommend remaking these figures in an appropriate manner for publication prior to Acceptance.

Experimental design

The methods for generating the web application are described in sufficient detail. I was able to download the data and code and launch the web application locally. Note that I received errors when running the app locally and as such was unable to view the data within the web application. However, I was able to access the data via the Shiny server without issue.

Validity of the findings

No comment.

Additional comments

I commend the authors for their well-developed web application. In addition, the manuscript is clearly written in professional, concise, and unambiguous language. As noted below, the weakness is in the presentation of the figures and in the lack of a clearly defined way to report bug issues with local instances of the web application, which should be improved upon before Acceptance.
1. Your figures need improvement to meet the standard of publication quality. Specifically, the figures appear to simply be screenshots of the Rstudio window with the WHONDRS-GUI running, resulting in low resolution images with ill-defined edges (i.e., you can clearly see the edges of background windows in the screenshots) and features that are not relevant to the image, such as the breadcrumb bar in Figures 1 and 3. It is also not clear what the inclusion of the scroll bar on the right-hand side of Figures 1 and 3 accomplishes in conveying information to the reader.
2. I thank the authors for providing the raw data and code, however the local instance of the web application resulted in errors when viewing the data in my browser. You should provide a clearly defined way of reporting bug issues to the code author(s). There was not an obvious way to do this through the ESS-DIVE archive, and there is no reference to a public repository (e.g., github) for the app code that would such built-in capabilities. Although the code is shared, this should be improved by providing end users with a way to report bugs with the code (preferably something more streamlined than simple email communication with the corresponding author, which I inferred as the only option for reporting this issue).

Reviewer 2 ·

Basic reporting

Integration of different types of knowledge worldwide about physical, chemical and biological processes in surface waters (rivers) to allow better predictions of possible scenarios of environmental perturbations is an interesting topic and a need

Experimental design

In my opinion, the more difficult aspect of the proposal is probably the integration of surface water metabolites and biological parameters information (which is highly changeable and dependent on time scale), with water physical properties and processes. Synchronization of both type of parameters is not easy.

Can the authors give more detail about how this crucial aspect can be approached?

Validity of the findings

Good proposal, interesting but challenging. Important aspects are not covered yet.

Additional comments

An important aspect not considered in the proposal is what data analysis tools should be used for the integration of information and for the acquisition of new knowledge, interpretation and prediction.

In the technology offered by the authors, only the access to the different type of data bases is given. This aspect is of great interest, but it is only the first step for the goal enunciated at the beginning of the paper, i.e. the understanding of the effects of river perturbations (dam operations, discharge, floods, draughts…) on the biogeochemistry of the river water resources. This should be made more clear in the paper.

---

## Round 0.2 · accepted · Accept

I am delighted to let you know that I have decided that you have satisfactorily addressed all reviewer comments. Hence, I am suggesting acceptance of the paper for publication in PeerJ. Thank you for cooperating with the review process.